# Inhibitory Effect of a Tankyrase Inhibitor on Mechanical Stress-Induced Protease Expression in Human Articular Chondrocytes

**DOI:** 10.3390/ijms25031443

**Published:** 2024-01-24

**Authors:** Yoshifumi Hotta, Keiichiro Nishida, Aki Yoshida, Yoshihisa Nasu, Ryuichi Nakahara, Shuichi Naniwa, Noriyuki Shimizu, Chinatsu Ichikawa, Deting Lin, Tomohiro Fujiwara, Toshifumi Ozaki

**Affiliations:** 1Department of Orthopaedic Surgery, Okayama University Graduate School of Medicine, Dentistry and Pharmaceutical Sciences, Okayama 700-8558, Japan; yoshifumi19870707@icloud.com (Y.H.); ayo@md.okayama-u.ac.jp (A.Y.); me19061@s.okayama-u.ac.jp (S.N.); me422049@s.okayama-u.ac.jp (N.S.); cntickw@icloud.com (C.I.); ldt773932009@gmail.com (D.L.); tozaki@md.okayama-u.ac.jp (T.O.); 2Locomotive Pain Center, Okayama University Hospital, Okayama 700-8558, Japan; 3Department of Orthopaedic Surgery, Okayama University Hospital, Okayama 700-8558, Japan; nasu_y@flute.ocn.ne.jp (Y.N.); pikumin55@gmail.com (R.N.); tomomedvn@okayama-u.ac.jp (T.F.)

**Keywords:** osteoarthritis, chondrocyte, mechanical stress, tankyrases, XAV939, SOX9, ADAMTS-5, MMP-13, IL-1β, NF-κB, β-catenin

## Abstract

We investigated the effects of a Tankyrase (TNKS-1/2) inhibitor on mechanical stress-induced gene expression in human chondrocytes and examined TNKS-1/2 expression in human osteoarthritis (OA) cartilage. Cells were seeded onto stretch chambers and incubated with or without a TNKS-1/2 inhibitor (XAV939) for 12 h. Uni-axial cyclic tensile strain (CTS) (0.5 Hz, 8% elongation, 30 min) was applied and the gene expression of type II collagen a1 chain (*COL2A1*), aggrecan (*ACAN*), SRY-box9 (*SOX9*), *TNKS-1/2*, a disintegrin and metalloproteinase with thrombospondin motifs-5 (*ADAMTS-5*), and matrix metalloproteinase-13 (*MMP-13*) were examined by real-time PCR. The expression of ADAMTS-5, MMP-13, nuclear translocation of nuclear factor-κB (NF-κB), and β-catenin were examined by immunocytochemistry and Western blotting. The concentration of IL-1β in the supernatant was examined by enzyme-linked immunosorbent assay (ELISA). TNKS-1/2 expression was assessed by immunohistochemistry in human OA cartilage obtained at the total knee arthroplasty. *TNKS-1/2* expression was increased after CTS. The expression of anabolic factors were decreased by CTS, however, these declines were abrogated by XAV939. XAV939 suppressed the CTS-induced expression of catabolic factors, the release of IL-1β, as well as the nuclear translocation of NF-κB and β-catenin. TNKS-1/2 expression increased in mild and moderate OA cartilage. Our results demonstrated that XAV939 suppressed mechanical stress-induced expression of catabolic proteases by the inhibition of NF-κB and activation of β-catenin, indicating that TNKS-1/2 expression might be associated with OA pathogenesis.

## 1. Introduction

Osteoarthritis (OA) is the most common joint disease [1] and the number of patients is expected to increase in the future. Although OA is a multifactorial disease, aging and obesity are likely to be major risk factors [2]. OA is a whole joint disease which affects multiple joints [3] and involves various processes, including articular cartilage degradation, osteophyte formation, synovitis, articular cartilage remodeling, meniscal degeneration [4], inflammation, and fibrosis of the infrapatellar fat pad [5]. A characteristic of OA is an imbalance between the repair and destruction of articular cartilage, and the disease involves complex inflammatory and metabolic factors. Although cartilage repair is promoted by chondrogenic differentiation and extracellular matrix (ECM) anabolism in normal chondrocytes [6], these functions are reduced in OA cartilage [7]. SRY-box9 (SOX9) has been identified as a master transcription factor of chondrogenesis in developing cartilage and has been found to control ECM homeostasis by regulating type II collagen (COL2A1) and aggrecan (ACAN) [8]. However, SOX9 activity is also downregulated and results in decreased ECM anabolism in OA cartilage [9]. On the other hand, excessive mechanical stress is involved in cartilage destruction and activates various inflammatory pathways, such as interleukin-1β (IL-1β), tumor necrosis factor-α, nuclear factor kappa-B, Wnt, transforming growth factor-β, and microRNA pathways, and oxidative stress. [10,11]. Excessive mechanical stress is associated with OA and affects both cartilage structure and the function of chondrocytes. Cartilage destruction is represented by hyaline cartilage deterioration, hypertrophic chondrocytes, the formation of cell clusters, and the loss of ECM [12]. Chondrocytes subjected to excessive stress release inflammatory cytokines such as IL-1β [13,14]. Inflammatory cytokines also produce matrix metalloproteinase-13 (MMP-13) [15] and a disintegrin and metalloproteinase with thrombospondin motifs-5 (ADAMTS-5) [16] which leads to degradation of ECM. Despite these insights into the pathogenesis of OA, the current treatment for OA is symptomatic and there is no curative treatment [17].

Tankyrases (TNKS-1 and TNKS-2) were recently reported as a potential target for disease-modifying osteoarthritis drugs (DMOADs) [18]. TNKS-1/2 belong to the poly (ADP-ribose) polymerase (PARP) superfamily which is involved in various cellular and molecular processes. PARPs transfer ADP-ribose to target proteins by post-translational modification, including poly ADP-ribosylation (PARylation) [19]. TNKS-1/2 are involved in various cellular functions, including telomere maintenance [20], Wnt signaling [21], glucose metabolism [22], and the heritable disease cherubism [23]. TNKS-1/2 are two homologous proteins and there is a slight structural difference; TNKS-1 has an His, Pro, and Ser (HPS) domain with unknown function [24]. The functions of TNKS-1/2 are considered to be similar due to their structural homology and protein binding properties [25]. A recent study has shown that TNKS-1/2-mediated PARylation of SOX9 plays an essential role in regulating SOX9 ubiquitination and degradation, and TNKS-1/2 inhibitors have a chondroprotective effect in a mouse OA model [26]. However, the involvement of TNKS-1/2 in the pathogenesis of OA in human chondrocytes and the mechanism of the chondroprotective effect of TNKS-1/2 inhibitors on a mechanical stress model remain poorly understood. This study aims to elucidate the involvement of TNKS-1/2 in OA pathogenesis and to investigate the effect of TNKS-1/2 inhibitors on mechanical stress-induced cartilage degeneration.

## 2. Results

### 2.1. The Effect of CTS Stimulation in Normal Human Chondrocytes

*COL2A1*, *ACAN*, and *SOX9* mRNA levels decreased at 12 h after CTS, which was significant only at 12 h after CTS and there were no significant differences at 0, 6, or 24 h after CTS compared with the control group (Figure 1A). *TNKS-1/2* were significantly increased at 12 h after CTS. *ADAMTS-5* and *MMP-13* were also significantly increased at 12 and 24 h after CTS (Figure 1B,C). Real-time PCR results showed that the CTS conditions (0.5 Hz, 8% elongation, 30 min) used in this study inhibited anabolic reactions and promoted catabolic reactions. In this study, we changed the elongation setting from 10% to 8%, and the same result was obtained as reported before [14,27,28,29]. Under these conditions, *TNKS-1/2* levels were upregulated over three-fold 12 h after CTS. This condition was found to increase the expression of *TNKS-1/2* the most and subsequent experiments were conducted using 8% elongation. The protein levels of *TNKS-1/2* were also increased after CTS (Figure 1D).

### 2.2. Effect of the TNKS-1/2 Inhibitor XAV939 on CTS-Induced Gene Expression and Concentration of IL-1β in the Supernatant in Normal Human Chondrocytes

We examined the effect of XAV939, a Tankyrase inhibitor, on the expression levels of *SOX9*, *COL2A1*, *ACAN*, *TNKS-1/2*, *MMP-13*, and *ADAMTS-5* after CTS. We examined the expression of the β-catenin gene (*CTNNB1*) at different concentrations by real-time PCR. The expression of *CTNNB1* was significantly downregulated in the 10 and 100 μM XAV939 treatment groups compared with the control group. However, the difference between the 10 and 100 μM groups was not significant. Based on these results, the concentration of XAV939 was determined to be 10 μM (Figure 2A). We determined the timing of sample detection in the presence of XAV939 to be 12 h after CTS, as the *TNKS-1/2* expression levels were most significantly increased at this time point. *SOX9*, *COL2A1*, and *ACAN* expression levels were reduced by CTS, whereas these decreases were blocked by XAV939. The expression levels tended to increase under the influence of XAV939 in situations without CTS (Figure 2B). In anabolic reactions, we found that XAV939 canceled the anabolic inhibition caused by CTS. The up-regulation of *MMP-13* and *ADAMTS-5* after CTS was canceled by XAV939 (Figure 2C). This result suggested that XAV939 inhibits the catabolic effect as well as promotes the anabolic effect in human chondrocytes. However, the expression of *TNKS-1/2* after CTS with XAV939 was not significantly decreased relative to those without XAV939 (Figure 2D). The concentration of IL-1β in the culture medium increased in a time-dependent manner after CTS in samples without XAV939. The concentration of IL-1β was increased most at 24 h after CTS and was not detected in groups without CTS. The upregulation of IL-1β was significantly decreased at 24 h after CTS combined with XAV939 treatment compared to those without XAV939 treatment (Figure 2E).

### 2.3. Effect of XAV939 on Activation of the MAPK Pathway in Normal Human Chondrocytes

Mechanical stress activates the stress response signaling pathway, MAPK, as reported before [14,27,29]. We therefore investigated MAPK (ERK, p38, and JNK) phosphorylation in the presence and absence of XAV939 at 30 min after CTS. Quantification of the percentage of ERK, p38, and JNK phosphorylation was performed (Figure 3). CTS significantly increased p38 and JNK phosphorylation compared with the non-stretched controls. The phosphorylation of ERK2 was significant but ERK1 was not in the CTS group compared with the control group (Appendix A). As shown by Western blot analysis, XAV939 did not inhibit the CTS-induced phosphorylation of MAPK.

### 2.4. The TNKS-1/2 Inhibitor XAV939 Suppressed CTS-Induced ADAMTS-5 and MMP-13 Protein Expression in Normal Human Chondrocytes

Both ADAMTS-5 and MMP-13 expression were upregulated and localized in the cytoplasm after CTS without XAV939, while expression was downregulated by treatment with XAV939 (Figure 4A). The percentages of chondrocytes positive for ADAMTS-5 and MMP-13 were significantly decreased by treatment with XAV939 (Figure 4B). A Western blot analysis and quantification of the protein levels were performed. These results showed that XAV939 inhibited the CTS-induced expression of ADAMTS-5 and MMP-13 (Figure 4C).

### 2.5. Effect of XAV939 on CTS-Induced Nuclear Translocation of NF-κB p65 and β-Catenin in Normal Human Chondrocytes

Immunocytochemistry showed that CTS induced NF-κB p65 and β-catenin translocation to the nucleus, but this was inhibited in the cells treated with XAV939 (Figure 5A). The percentages of chondrocytes positive for the nuclear translocation of NF-κB p65 and β-catenin were significantly decreased by treatment with XAV939 (Appendix A). The result of Western blot analysis of the nuclear extraction proteins also showed that XAV939 inhibited the CTS-induced protein expression levels of NF-κB p65 and β-catenin (Figure 5B). This result suggested that Wnt/β-catenin signal activation also occurred in this mechanical stress model.

### 2.6. Double Immunocytochemical Staining of NF-κB p65 and β-Catenin in Normal Human Chondrocytes

Immunocytochemistry showed that NF-κB p65 and β-catenin translocation to the nucleus occurred simultaneously after CTS and this effect was blocked in the presence of XAV939. The percentages of chondrocytes exhibiting the nuclear translocation of NF-κB p65 and β-catenin were significantly decreased by treatment with XAV939. Quantification in the form of nuclear β-catenin-positive cells/nuclear NF-κB p65-positive cells was performed (Appendix A). In samples after CTS without XAV939, all cells with NF-κB p65 translocated to the cell nucleus also showed β-catenin nuclear translocation. This result suggested that NF-κB and β-catenin nuclear translocation interacted with each other and might affect subsequent protease expression (Figure 6A,B).

### 2.7. Immunohistochemical Evaluation of TNKS-1/2 in Human Articular Cartilage Tissues

Immunohistochemical staining was carried out using 18 OA cartilage samples from 14 patients to evaluate the expression of TNKS-1/2 in articular cartilage. The extent of cartilage destruction in each specimen was evaluated using the Mankin score [30]. In specimens with low-grade cartilage degradation, TNKS-1/2 were rarely expressed in every layer. In those with mild-grade degradation, TNKS-1/2 expression in the superficial layer was significantly increased relative to the middle and deep layers. In moderate-grade cartilage, TNKS-1/2 expression was also significantly increased in the middle and deep layers compared to the superficial layer. Although decreased compared to other grades, TNKS-1/2 expression was identified in the middle or deep layers in severe grades (Figure 7A,B).

A significantly higher expression of TNKS-1/2 was seen in the superficial layer of mild-grade degradation rather than in other layers, and in the deep layer of moderate-grade degradation rather than in the superficial layer (Figure 7C).

## 3. Discussion

In the current study, we demonstrated the importance of TNKS-1/2 as a factor in mechanical stress-induced cartilage degeneration in normal human chondrocytes in vitro. Our data also suggested that mechanical stimulation up-regulated the expression of TNKS-1/2 and that the nuclear translocation of β-catenin by activation of Wnt/β-catenin signaling might enhance NF-κB–IL-1β signaling, resulting in the production of ADAMTS-5 and MMP-13. We also found that intervention by XAV939, a TNKS-1/2 inhibitor, suppressed these pathways and promoted an anabolic reaction. In human OA cartilage, TNKS-1/2 expression was seen more frequently in mild- and moderate-grade OA cartilage when compared with low- and severe-grade OA cartilage.

We have previously reported that CTS (0.5 Hz, 10% elongation, 30 min) induced the expression of MMP, ADAMTS, and other proteases in normal human chondrocytes. RUNX2 and IL-1β play an important role in these catabolic reactions through MAPK and NF-κB [14,27,28,29]. In this study, we investigated the conditions in which *TNKS-1/2* expression was significantly upregulated compared with the control group, and CTS with elongation set at 8% was selected. Although *MMP-13* was also increased at 10% elongation, the reason for the reduced expression of *TNKS-1/2* at 10% elongation compared to 8% elongation group was not clear (Appendix A). Chondrocytes are usually exposed to a combination of different forces such as compression, tension, and shear force [31]. However, it is not clear what kind of loading affects catabolic and anabolic reactions in chondrocytes. In the case of chondrocytes subjected to CTS using a Flexcell strain unit (FX2000, FX3000, FX4000, and FX5000), it has been reported that loading cells with up to 3% strain, at 0.17 Hz, for 2 h, resulted in weak or no biological responses, loading between 3–10% strain, 0.17–0.5 Hz, and 2–12 h led to anabolic responses, while above 10% strain, at greater than 0.5 Hz, for more than 12 h, catabolic events predominated [32]. Although results indicated that the CTS conditions applied in this study promoted catabolic reactions, the differences from the previous report are due to the following reasons. First, the form of mechanical strain is biaxial in the reference, whereas uniaxial in our study. Second, the type of cells is from normal animals in the reference, whereas from normal humans in our study.

In anabolic reactions, the expression levels of *COL2A1*, *SOX9*, and *ACAN* were up-regulated by XAV939 treatment in human chondrocytes, which is similar to the report showing that TNKS-1/2 inhibition resulted in cartilage protection through the suppression of SOX9 degradation in mouse cartilage [26]. The reason why SOX9 was not significant but tended to upregulate at 24 h after CTS might be explained by the report that SOX9 reactively upregulated in early OA [33]. The increase in anabolic factors seen at 24 h after CTS might be due to the compensatory effect against catabolic stimulation by CTS, the longer incubation time after mechanical stimulation, and the use of chondrocytes harvested from relatively younger patients with high cartilage repair ability in this study.

Among the catabolic reactions, the expression of ADAMTS-5 and MMP-13 increased over time after CTS. In this study, we found that the NF-κB–IL-1β and Wnt/β-catenin pathways were both activated and involved in these reactions. It is known that the transcription of NF-κB plays an important role in the production of ADAMTS-5 [34] and MMP-13 [35] and that NF-κB induces the production of IL-1β. IL-1β is also known to induce the production of NF-κB [36]. NF-κB nuclear translocation was observed in the early phase after CTS by immunocytochemistry, and the concentration of IL-1β in the cell culture medium, evaluated by ELISA, was increased in the late phase after CTS. This suggested that the production of IL-1β by NF-κB signaling required some time and that the interaction between IL-1β and NF-κB synergistically increased the expression of ADAMTS-5 and MMP-13. As for the Wnt/β-catenin pathway, it has been reported that the increase in β-catenin leads to matrix degradation and the suppression of chondrocyte proliferation [37]. As previously reported, Wnt/β-catenin might play important roles in the up-regulation of ADAMTS-5 and MMP-13 in chondrocytes [38,39].

In the current study, the expression of TNKS-1/2 increased over time after CTS, similar to ADAMTS-5 and MMP-13, suggesting the involvement of TNKS-1/2 in the catabolic reaction. TNKS-1/2 are similar in structure with 81% nucleotide homology and 85% amino acid identity. TNKS-1/2 also share most of their functionality [40] and it has been shown that TNKS-1/2 are necessary to maintain some specific functions such as telomere cohesion, telomere length regulation, mitotic spindle integrity, and protein degradation [41]. The expression of TNKS-1/2 was almost the same and no differences were observed throughout the study. TNKS-1/2 promotes Wnt/β-catenin signaling by the poly(ADP-ribosyl)ation of AXIN, a negative regulator of Wnt/β-catenin, which leads to ubiquitin degradation [42]. As a result, the intracellular accumulation and nuclear translocation of β-catenin promote the expression of its target molecules.

TNKS-1/2 have been investigated as a therapeutic target for the Wnt signaling pathway-dependent cancers [43,44]. The importance and usefulness of many TNKS-1/2 inhibitors have been reported and have resulted in the development of many TNKS-1/2 inhibitors including XAV939, IWR-1, G007-LK, JW55, AZ1366, JW74, and NVP-TNKS656 [45]. Among these, XAV939 and IWR-1 showed chondroprotective effects in a mouse OA model [26]. IWR-1 was first reported as a TNKS-1/2 inhibitor in 2009 [46] followed by the identification of XAV939, which functions similarly to IWR-1 [42]. XAV939 and IWR-1 were used in a paper reporting the chondroprotective effects of Tankyrase inhibitors and the effects were described as similar [26] and the use of XAV939 on human chondrocytes was reported before [47]. Therefore, we decided to use XAV939 as a TNKS-1/2 inhibitor in this study. Regarding the concentration of XAV939, 10 μM was reported to be effective in inhibiting Wnt/β-catenin signaling in chondrocytes [47] and the same results were observed in our experiments, so we used this concentration. Treatment with XAV939 suppressed the expression of ADAMTS-5 and MMP-13 as well as the increase in IL-1β concentration in the culture medium. This result implied that XAV939 inhibited the nuclear translocation of NF-κB and the subsequent increase in IL-1β concentration and consequently suppressed the expression of ADAMTS-5 and MMP-13. In addition, CTS promoted nuclear translocation of β-catenin and indicated activation of the Wnt/β-catenin pathway in CTS as well as NF-κB. More importantly, XAV939 treatment significantly suppressed the nuclear translocation of β-catenin and NF-κB simultaneously. These results suggest that TNKS-1/2 inhibitors exert their chondroprotective effects through the interaction between Wnt/β-catenin signaling and the NF-κB–IL-1β pathway.

The relationships between Wnt/β-catenin signaling and NF-κB have been discussed, but their interrelation is still controversial. The regulatory effects of Wnt/β-catenin signaling on the NF-κB pathway in chondrocytes have been reported as both negative [48] and positive [49,50,51]. On the other hand, the regulatory effects of the NF-κB pathway on Wnt/β-catenin signaling in chondrocytes are reported to be positive regulations [52]. In this study, nuclear translocation of β-catenin and NF-κB occurred simultaneously after CTS without XAV939, while they were both suppressed after CTS with XAV939. This suggested that Wnt/β-catenin signaling had a positive effect on NF-κB activity and led to the subsequent expression of ADAMTS-5 and MMP-13.

The MAPK signaling pathway is activated by multiple internal and external stimulants, such as extracellular signals, physical stimulation, bacteria, tumor necrosis factor (TNF), and inflammatory cytokines like IL-1 and IL-6 [53]. It is established that JNK, ERK, and p38 in the MAPK family are involved in OA pathology and that p38 and JNK are necessary for IL-1 induction of MMP-13 in chondrocytes [35]. In our results, phosphorylation of p38 and JNK were increased relative to control groups after CTS and these effects were not inhibited by XAV939. Crosstalk between the MAPK pathway and Wnt signals has an important role in cancer, and their importance has also been reported in cartilage regeneration [54]. Some studies have shown that activation of Wnt5a signaling by IL-1β induced the expression of MMPs via the JNK pathways in rabbit temporomandibular joint (TMJ) condylar chondrocytes [55]. As Wnt5a is a non-canonical Wnt signal, XAV939 inhibition of β-catenin-dependent canonical Wnt3a was presumed to not affect MAPK pathways.

The genetic relationship between TNKS-1/2 and OA has been reported previously [56,57]. Histological evaluation of TNKS-1/2 expression in human cartilage showed that TNKS-1/2 were expressed in damaged cartilage tissues and were not found in normal cartilage [26]. However, the relationship between cartilage degradation grade and TNKS-1/2 expression in human cartilage tissues has not been investigated. In human OA cartilage tissues, TNKS-1/2 expression was increased most in the superficial layer in mild-grade OA, which was similar to the localization of ADAMTS-5 and MMP-13 in human OA cartilage [58]. In addition, TNKS-1/2 expression was seen more frequently in mild and moderate-grade OA cartilage when compared with severe-grade OA cartilage. This result was consistent with the previous report that the expression of MMP-13 was increased in grade II OA (Mankin score 5–9) and decreased in grade III OA (Mankin score 10–14) in human OA cartilage [59]. Moreover, TNKS-1/2 were also expressed in the formation of clusters, which are specific for the progression of OA [60]. These results suggested that TNKS-1/2 might be involved in OA pathology in human cartilage tissues.

There were several limitations to this study. First, cells were cultured in monolayers and the stretch machine was used as a simple model for OA in this study. During normal movement, mechanical stress, such as compressive, elongation, and shear stress, is applied to chondrocytes in cartilage and the results might be different; consequently, other kinds of mechanical stress should be investigated. The second limitation was the number of clinical samples, as we were unable to collect more samples within the period.

## 4. Materials and Methods

### 4.1. Cells and Cell Culture

Normal human articular chondrocytes derived from knee cells (NHAC-kn) obtained from a 26-year-old male and a 30-year-old male were purchased from Lonza (Walkersville, MD, USA). Cells were cultured at 37 °C in chondrocyte basal medium (CBM, Lonza, Walkersville, MD, USA) containing growth medium, fetal bovine serum (FBS), transforming growth factor-beta (TGF-β), R3 insulin-like growth factor (R3-IGF), transferrin, insulin, gentamicin, and amphotericin-B (CDM™ BulletKit, Lonza). The attached cells were incubated at 37 °C in 5% CO_2_ in a humidified atmosphere and subcultured on type I collagen-coated polystyrene tissue culture dishes (Iwaki, Shizuoka, Japan). NHAC-kn was used in passage 3.

### 4.2. Cyclic Tensile Strain

Human chondrocytes were seeded onto stretch chambers coated with type I collagen at a concentration of 1 × 10^5^ cells/chamber. Each chamber had a culture surface of 2 × 2 cm. After 12 h, the cells were attached to the chamber. Mechanical stress was applied using the ST-140-10 mechanical stretch system (STREX, Osaka, Japan). The end of the chamber is attached to a metal frame and the opposite end is held on a movable frame that is connected to a motor-driven shaft. This apparatus allows the entire silicon membrane to be stretched uniformly [61]. In our previous study, CTS (0.5 Hz, 10% elongation) was applied for 30 min to cause catabolic stress. CTS (0.5 Hz, 8% elongation) was applied for 30 min for all experiments in this study. The CTS elongation rate was changed to 8% because the catabolic reaction was promoted and the expression of *TNKS-1/2* was most up-regulated in this condition (Appendix A). Cells without mechanical stress were seeded into the same chambers and used as controls.

### 4.3. Treatment with TNKS-1/2 Inhibitor

TNKS-1/2 inhibitor (XAV939) was purchased from Sigma-Aldrich (St. Louis, MO, USA), dissolved in dimethyl sulfoxide (DMSO), and then diluted to 10 μM. Cultures were treated with XAV939 for 12 h before application of CTS.

### 4.4. ELISA for IL-1β in Culture Medium

Cell culture supernatants were collected at 0, 6, 12, and 24 h after CTS. The volume of culture supernatant collected from each well was 2 mL. Culture supernatants were immediately centrifuged at 1500× *g* for 10 min at 4 °C to remove cells and were stored at −80 °C until examination. The concentration of IL-1β in the supernatant was measured using a highly sensitive IL-1β enzyme-linked immunosorbent assay (ELISA) kit (Quantikine HS ELISA Human IL-1β/IL-1F2 Immunoassay, R&D Systems, Inc., Minneapolis, MN, USA), according to the manufacturer’s protocol. The absorbance was measured by iMark Microplate Reader (Bio-Rad, Richmond, CA, USA).

### 4.5. Quantitative Real-Time PCR Analysis

The cells were washed twice with PBS and total RNA was isolated using an RNeasy mini kit (Qiagen, Hilden, Germany) according to the manufacturer’s protocol. RNA samples (500 ng) were reverse transcribed using Primescript RT master mix (Takara Bio Inc., Shiga, Japan). The resulting cDNAs were used for real-time polymerase chain reaction (PCR) amplification. Real-time PCR was performed using the QuantStudio™ 1 Real-Time PCR system (Thermo Fisher Scientific, Waltham, MA, USA) with TaqMan Gene Expression Assays for human *COL2A1* (Hs002645051_m1), *ACAN* (Hs00153936_m1), *SOX9* (Hs00165814_m1), *TNKS-1* (Hs00186671_m1), *TNKS-2* (Hs00228829_m1), *ADAMTS-5* (Hs01095518_m1), *MMP-13* (Hs00942584_m1), *CTNNB1* (Hs00355045_m1), and GAPDH (Hs02786624_g1) (Applied Biosystems, Foster City, CA, USA). Application of a housekeeping gene, GAPDH, was used for normalizing the efficiency of cDNA synthesis and the amount of RNA. We calculated the final expression levels by dividing the expression level of *COL2A1*, *ACAN*, *SOX9*, *TNKS-1*, *TNKS-2*, *ADAMTS-5*, *MMP-13*, and *CTNNB1* by the expression level of *GAPDH*. Each value obtained for the control cells (un-stretched cells without XAV939) was set to 1.

### 4.6. Immunocytochemistry

Immunocytochemistry was used to observe the mechanical stress-induced expression and localization of ADAMTS-5, MMP-13, β-catenin, and NF-κB p65 proteins. Cells were stretched with or without XAV939 pretreatment according to the protocols described above, then fixed with 1% paraformaldehyde solution at 24 h after application of CTS for ADAMTS-5, and MMP13, and at 30 min after application of CTS for β-catenin and NF-κB p65. The membranes of the culture chambers were then removed and incubated with anti-ADAMTS-5 antibody (1:100, ab45042, Abcam, Cambridge, MA, USA), anti-MMP-13 antibody (1:100, ab39012, Abcam), anti-β-catenin antibody (1:3200, 15B8, Cell Signaling Technology, Danvers, MA, USA), or anti-NF-kB p65 antibody (1:400, D14E12, Cell Signaling Technology) for 120 min at room temperature. Bovine serum albumin-containing solutions without primary antibodies were used as negative controls. We used Alexa Fluor 488-conjugated antibody (anti-mouse/rabbit) as the secondary antibody, and Hoechst 33342 (ICN Biomedicals, Aurora, OH, USA) for nuclear staining. Samples were observed under a fluorescence microscope (Leica Microsystems Ltd., Wetzlar, Germany), and protein expression was evaluated by the proportion of cells positive for ADAMTS-5, MMP-13, β-catenin, and NF-kB p65 (number of positive cells/all cells). The cell number was counted in four fields, at 100× magnification for NF-κB p65 and β-catenin nuclear translocation and 10× magnification for ADAMTS-5 and MMP-13, and the mean value from four fields was calculated.

### 4.7. Western Blot Analysis

All cells stretched on the chambers were resuspended in Mammalian Protein Extraction Buffer (GE Healthcare, Piscataway, NJ, USA) and proteins were extracted. The nuclear extracts for analysis of NF-kB p65 and β-catenin were collected using EPIXTRACT Nuclear Protein Isolation Kit II (Enzo Life Sciences, Farmingdale, NY, USA), according to the manufacturer’s protocol. Each concentration of the protein was measured using a Bradford Protein Assay Kit (TaKaRa Bio Inc.) and was adjusted in each sample. Samples (10 μg of total protein/lane, 15 μg/lane for NF-kB p65 and β-catenin) were loaded onto gels for sodium dodecyl sulfate-polyacrylamide gel electrophoresis (SDS-PAGE) using a Mini-Protean^®^ Tris-glycine extended gel (Bio-Rad, Richmond, CA, USA) for 1 h at 100 V before being transferred to PVDF membranes. The membranes were incubated with blocking reagent (Toyobo, Shiga, Japan) for 1 h at room temperature and incubated overnight at 4 °C with primary antibodies to ERK1/2, JNK, p38 MAPK (dilution, 1:1000 Cell Signaling Technology), phosphorylated ERK1/2 (Thr202/Tyr204), JNK (Thr183/Tyr185), p38 (Thr180/Tyr182) (dilution, 1:1000 Cell Signaling Technology), TNKS-1/2 (dilution, 1:1000; cat. no. sc-365897; Santa Cruz Biotechnology, Santa Cruz, CA, USA), ADAMTS-5 (1:500, ab45042, Abcam), MMP-13 (1:3000, ab39012, Abcam), anti-β-catenin (1:1000, 15B8, Cell Signaling Technology), or anti-NF-kB p65 (1:1000, D14E12, Cell Signaling Technology). After washing, the membranes were incubated with IRDye Goat anti-rabbit IgG (LI-COR Biosciences, Lincoln, NE, USA) or IRDye Goat anti-mouse IgG (LI-COR Biosciences) as secondary antibodies for 1 h at room temperature. Immunoreactive proteins were detected using the OdysseyFc Imaging System (LI-COR Biosciences). Original, uncropped western blot images were shown in Appendix A.

### 4.8. Clinical Samples of Human Cartilage Tissue

Clinical samples were obtained from 14 patients (18 samples) who were diagnosed with OA and who underwent total joint replacement of the knee at our hospital (Okayama University Hospital, Okayama, Japan). Cartilage samples from patients with primary OA were included. Samples with OA secondary to significant trauma, congenital joint abnormalities, metabolic disorders, infections, endocrine disorders, neuropathic diseases, and rheumatological or other systemic diseases were excluded. The current study received the approval of the Ethics Committee of Okayama University Graduate School of Medicine, Dentistry, and Pharmaceutical Sciences (2304-002; approval date; 10 March 2023). Individual patient data included age, gender, body mass index (BMI), and Kellgren–Lawrence (KL) grade. The patients comprised 3 men and 11 women aged 52–85 years at the time of the operation (Appendix A).

Articular cartilage samples were fixed in 10% formalin solution and then decalcified in 20% ethylenediaminetetraacetic acid (EDTA) before embedding in paraffin blocks.

### 4.9. Histological Evaluation of Cartilage Destruction

Safranin O–fast green stain was performed on articular cartilage, and we evaluated the severity of joint destruction in each joint using the histologic histochemical grading system described by Mankin et al. [62]. The Mankin system assesses four parameters, including the structure of articular cartilage (0–6), the cellularity of chondrocytes (0–3), proteoglycan staining (0–4), and the irregularity of tide marks (0–1). The scores are summed to determine a total score ranging from normal cartilage (Score 0) to severe degeneration (Score 14). The OA severity of the cartilage samples was defined as low (Mankin score 0–2), mild (Mankin score 3–6), moderate (Mankin score 7–10), or severe (Mankin score 11–14).

### 4.10. Immunohistochemical Evaluation of TNKS-1/2 in Articular Cartilage and the Expression Ratio in Chondrocytes

Deparaffinized cartilage sections were pretreated with 0.01 M citrate buffer (pH 6.0) in an autoclave at 90 °C for 10 min to retrieve the antigen. Mouse anti-TNKS-1/2 antibodies (dilution, 1:50; cat. no. sc-365897; Santa Cruz Biotechnology) were used as the primary antibody and were incubated with the samples at 4 °C overnight. Histofine1 Simple Stain Rat MAX PO (M) (Nichirei Biosciences, Tokyo, Japan) was used as the secondary antibody. The reaction was visualized by diaminobenzidine, and counterstaining was carried out with hematoxylin. Sections incubated with non-immune mouse serum were used as negative controls. TNKS-1/2 expression in articular cartilage was evaluated in superficial, middle, and deep layers, and the proportion of positive cells was determined by dividing the number of chondrocytes positive for TNKS-1/2 by the total number of chondrocytes. The value was the mean of three fields of view. The magnification when obtaining images was set at ×100 to cover the entire layer of articular cartilage.

### 4.11. Statistical Analysis

All data are expressed as the mean ± 95% confidence intervals (CI). Differences among individual sample groups were analyzed using a one-way analysis of variance (ANOVA) with Tukey’s multiple comparisons tests. Differences between two groups were compared by paired *t*-test. All analyses were conducted using GraphPad Prism 9 (GraphPad Software, San Diego, CA, USA) with a *p*-value < 0.05 regarded as significant.

## 5. Conclusions

Our study showed that the inhibition of the TNKS-1/2-mediated synergistic activation of the Wnt/β-catenin and NF-κB pathways decreased the expression of ADAMTS-5 and MMP-13 and resulted in chondroprotective effects in vitro. We also found that TNKS-1/2 expression in human OA cartilage tissues was related to cartilage degeneration. These results cumulatively suggested that TNKS-1/2 could be a target of molecular treatment and a diagnostic marker for OA.

## Figures and Tables

**Figure 1 ijms-25-01443-f001:**
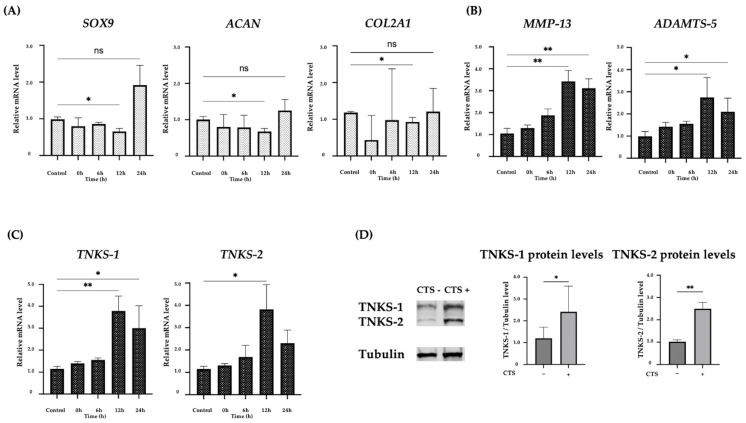
(**A**) Expression of anabolic factors after cyclic tensile strain in normal human chondrocytes. The expression of each factor decreased up to 12 h after the application of CTS. (**B**) Expression of catabolic factors after CTS. The peak expression was at 12 h after CTS. (**C**) The expression of *TNKS-1/2* after CTS. The peak expression was at 12 h after CTS. (**D**) The increase in TNKS-1/2 protein levels after CTS by Western blot analysis and quantification of TNKS-1/2 protein levels (* *p* < 0.05, ** *p* < 0.01, ns: not significant).

**Figure 2 ijms-25-01443-f002:**
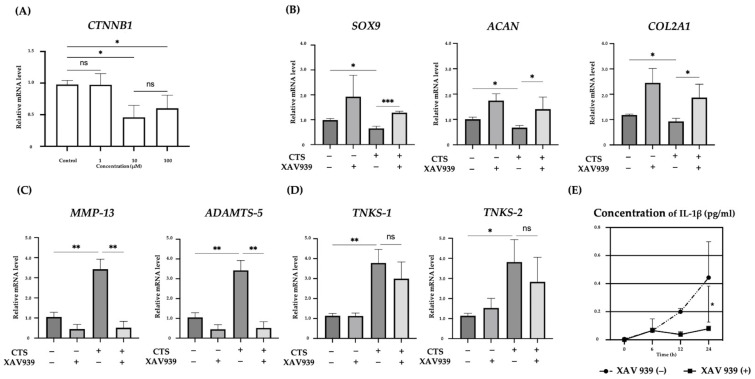
(**A**) Relative expression of *CTNNB1* mRNA in normal human chondrocyte cells, treated with different XAV939 concentrations (0, 1, 10, and 100 μM). (**B**–**D**) The effects of a TNKS-1/2 inhibitor (XAV939: 10 μM) on the expression of anabolic (**B**), and catabolic factors (**C**) and TNKS-1/2 (**D**) in human chondrocytes at 12 h after CTS. (**E**) The concentration of IL-1β in culture medium after CTS with or without XAV939 (* *p* < 0.05, ** *p* < 0.01, *** *p* < 0.0001, ns: not significant).

**Figure 3 ijms-25-01443-f003:**
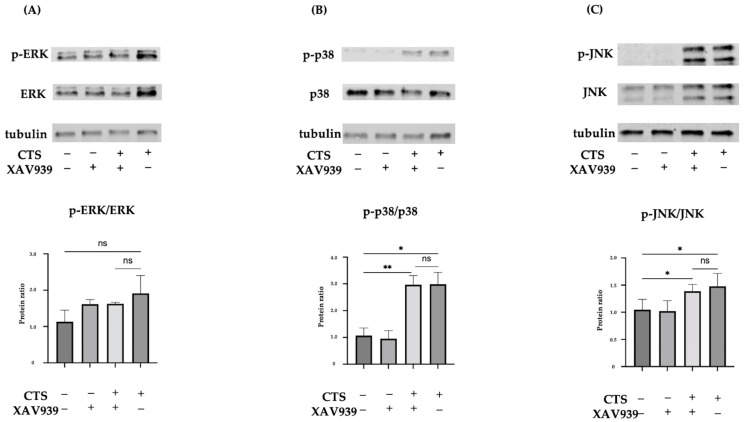
Western blots for total ERK and phosphorylated ERK (p-ERK) (**A**), total p38 and phosphorylated p38 (p-p38) (**B**), or total JNK and phosphorylated JNK (p-JNK) (**C**) in normal human chondrocytes (* *p* < 0.05, ** *p* < 0.01, ns: not significant).

**Figure 4 ijms-25-01443-f004:**
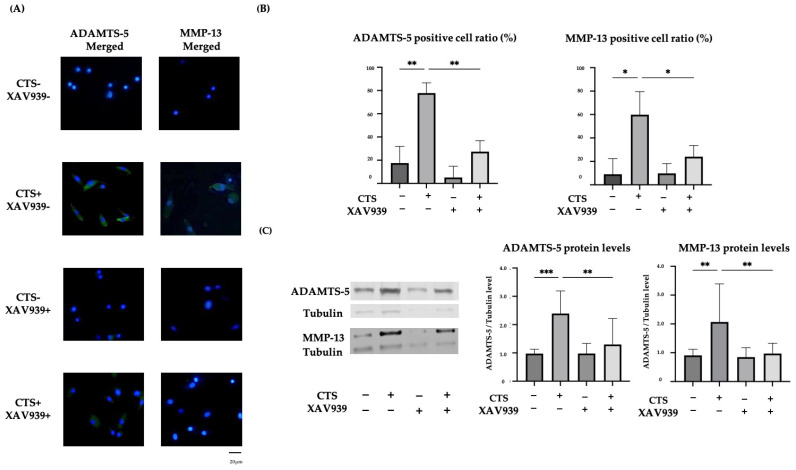
(**A**) The effect of mechanical stress and XAV939 on the expression of ADAMTS-5 and MMP-13 in normal human chondrocytes. Up-regulation of ADAMTS-5 and MMP-13 was localized to the cytoplasm (DAPI: blue signal, ADAMTS-5 and MMP-13: green signal). Scale bar = 20 μm. (**B**) The percentages of chondrocytes positive for ADAMTS-5 and MMP-13. Cell numbers were counted in four fields, at 10× magnification, and the mean was calculated (* *p* < 0.05, ** *p* < 0.01, *** *p* < 0.0001). (**C**) The effect of XAV939 on the protein expression of ADAMTS-5 and MMP-13 by Western blot and quantification analysis.

**Figure 5 ijms-25-01443-f005:**
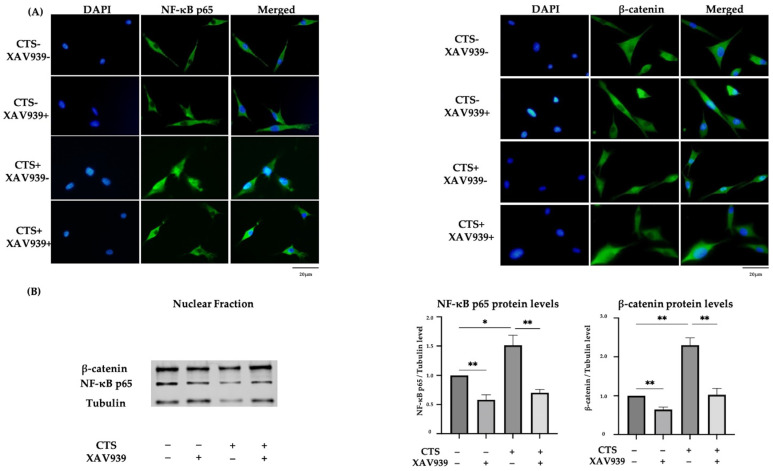
(**A**) The effect of mechanical stress and XAV939 on nuclear translocation of NF-κB p65 and β-catenin by immunocytochemistry in normal human chondrocytes (DAPI: blue signal, NF-κB p65 and β-catenin: green signal). Scale bar = 20 μm. (**B**) Western blot analysis and quantification of nuclear extraction proteins for NF-κB p65 and β-catenin. (* *p* < 0.05, ** *p* < 0.01).

**Figure 6 ijms-25-01443-f006:**
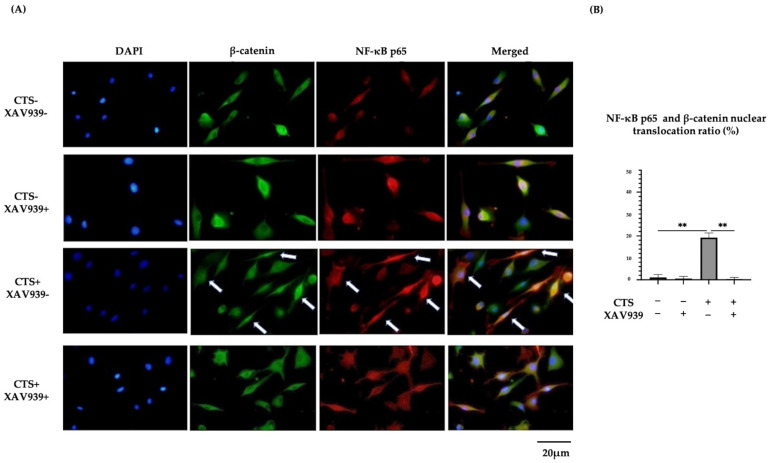
(**A**) Double immunohistochemical staining of β-catenin and NF-κB p65 nuclear translocation in normal human chondrocytes (DAPI: blue signal, β-catenin: green signal, NF-κB p65: red signal). Arrows showed double staining cells. (**B**) Both β-catenin and NF-κB p65 translocation to nuclei were observed after CTS without XAV939, while this effect was canceled by XAV939 (** *p* < 0.01).

**Figure 7 ijms-25-01443-f007:**
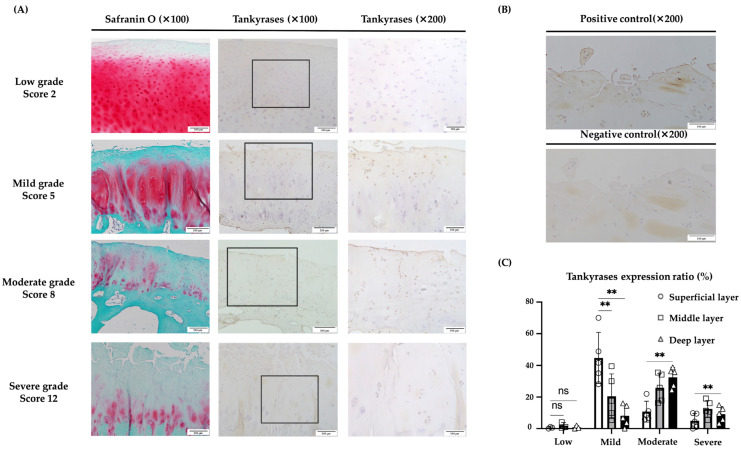
Immunohistochemical evaluation of TNKS-1/2 in human articular cartilage tissues. (**A**) In tissues with low-grade OA (Mankin score 0–2), TNKS-1/2 expression was not clear. In tissue with mild-grade OA (Mankin score 3–6), the main expression of TNKS-1/2 was in superficial and deep layers. In tissue with moderate-grade OA (Mankin score 7–10), the main expression of TNKS-1/2 was in the middle to deep layers. In tissue with severe-grade OA (Mankin score 11–14), TNKS-1/2 expression was observed in the deep layer but not in the superficial and middle layers. (**B**) No signal was observed in the negative control. (**C**) In low-grade OA, TNKS-1/2 were rarely expressed in every layer. In mild-grade OA, TNKS-1/2 expression was significantly increased in the superficial layer relative to the middle and deep layers. In moderate-grade OA, expression was significantly increased in the deep and middle layers compared to the superficial layer. In severe-grade OA, the expression was low relative to other grades and the difference between layers was not clear (** *p* < 0.01, ns: not significant). The scale bars were set at 200 μm at 100× magnification and 100 μm at 200× magnification.

## Data Availability

Data contained within the article.

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
