# Peer review of "Inhibitory Effect of a Tankyrase Inhibitor on Mechanical Stress-Induced Protease Expression in Human Articular Chondrocytes"

_ijms, 2024, doi:10.3390/ijms25031443_

Round 1
Reviewer 1 Report (New Reviewer)
Comments and Suggestions for Authors
Major comments
1. In Fig. 1A, why do the expressions of tankyrases show only transient difference in 12 h post-CTS? Why do the SOX9 and ACAN levels rather increase at 24 h time point, and how do the authors expect this trend would go at the longer periods?
2. What is the authors' explanation for shifting of TNKS-positivity toward deep layer as OA progresses (line 197-8)?
3. What is the authors' explanation for more dramatic increase of TNKS expressions under 8% CTS, than 10% (line 223-5)?
4. Statistical analysis should be reconducted. It is unclear whether the authors underwent one-way or two-way ANOVA test prior to Tukey's post-hoc test.
Minor comments
1. Western blot analysis data in Fig. 3 should be supported by quantification with multiple biological replicates. In line 135, the data explanation for Fig. 3A should be more elaborated.
2. Fig. 4C, the western blot quality is low for ADAMTS-5. This data should also be quantified. Throughout the manuscript, separate gels should be spaced (or separated) to aviod confusion.
3. Fig. 5A and Fig. 5B should be quantified. Fig. 5B and D, the same tubulin blot is used for different figures. No y-axis scale for Fig. 5B. The band thickness for tubulin blot is not equal among treatment groups. This may cause bias for its quantification graph.
4. In line 178, this statement can only be valid after the IF quantification in the from of (% nuclear b-catenin-positive cells / nuclear p65-positive cells).
5. Why is there only 15 samples in Fig. 7, unlike the total number of obtained samples (line 427)?
Comments on reference usage
- Ref. 8 in line 61 should be supported with other study that covers the effect of mechanical stress on changes in cartilage.
- Ref. 9 in line 63 would better be replaced with original research article, not review.
- Ref. 22 in line 190 is for Mankin scoring of hip OA. The authors should replace with the knee one in alignment with what they used in this study.
- Line 220-1 should be supported with reference article.
- Ref. 33 in line 263 requires more description.
Comments on the Quality of English Language1. typos
- Line 24 and 216, TNKS-1/2
- Line 246, IL-1b
- Line 272, JW74
- Line 287, 386, NF-kB
2. expressions
- Line 82, mark as SOX9 "mRNA" levels
- A sentence ranging line 292-3 is confusing.
- Line 312, The "genetic" relationship
Author Response
Please see the attachment.

Reviewer 2 Report (New Reviewer)
Comments and Suggestions for Authors
My comments are as follows:
The abstract is very long. It should be summarized.
Introduction
Lines 50-52: It should be explained that OA is a whole joint disease involving all joint tissues, including meniscal degeneration, inflammation and fibrosis of infrapatellar fat pad and subchondral bone remodeling.
Abbreviations should be defined at first mention such as SOX9.
Lines 59-63: this part should be expanded. Considering the aim of the manuscript, the biomechanical changes of OA cartilage and chondrocytes should be better described (DOI:10.3390/biomedicines11071942, DOI:10.3390/pr11041014 etc).
It should be better explained the difference between TNKS-1 and 2 and the their function.
Materials and methods
Section 4.1.: it is unclear to me if the authors maintained separated the two primary cell lines and the experiments were performed in triplicate using these chondrocytes (for a total of six).
Lines 358-359: how did the authors select the concentration of the inhibitor to use?
Section 4.4.: Plate reader should be added. Genes names should be written in italics.
The authors should check all the manuscript and write in italics all gene names.
Lines 410-412: how many micrograms of protein were loaded on SDS-PAGE?
Section 4.8: criteria of patients inclusion/Exclusion should be added. Did the patients sign an informed consent? Ethical approval protocol and date should be added.
Line 438: Mankin et al. Reference should be added.
Lines 461: did the authors use ANOVA followed by Tukey comparison?
Results and discussion
Figures: following journal guidelines, authors should use lowercase letters written at the bottom center of each figure section.
Figure 1d: I can see two bands on the western blot of TNKS-1/2. I suppose that these two bands are TNKS-1 and TNSK-2. If yes, could the authors mark the two on the blot?
Line 101: could the authors describe XAV939?
Figure 2E: x-axis is missing.
Figure 3: the authors should quantify the western blot bands reporting the related graphs with statistical analysis. All the western blots should be quantified.
Figure 4a is unclear. Did the authors use DAPI? Could the authors enlarge the images in order to better see the cells?
Figure 7: histological images should be enlarged.
Figure 7C: the authors reported mild, moderate and severe OA sample, low grade samples are missing.
Lines 197-198 and lines 217-219: this part should be better explained. It seems not to be supported by the data reported in figure 7.
The data presented in figure 7 are confused. It is not clear the role of TNKS-1/2 as OA progresses. It should be better clarified. Other experiments/data should be reported to clarify this part.
Lines 275-276: why did the authors select this inhibitor?
Lines 328-329: here the authors reported that stimulated the cells with IL-1b. Is that correct? In which experiments?
Round 2
Reviewer 1 Report (New Reviewer)
Comments and Suggestions for Authors
Supplemental Figures should be labeled in the order of their appearance in the manuscript.
Comments on the Quality of English LanguageTNKS-1/2 (line 21)
10 uM (line 111)
analysis, (line 141)
why was b-catenin italicized in some cases?
min (line 233, 384, 477)
Author Response
Please see the attachment.

Reviewer 2 Report (New Reviewer)
Comments and Suggestions for Authors
No other comments
Author Response
Please see the attachment.

This manuscript is a resubmission of an earlier submission. The following is a list of the peer review reports and author responses from that submission.
Round 1
Reviewer 1 Report
Comments and Suggestions for Authors
The authors investigated the role and mechanisms of TNKS in the pathogenesis of OA in human chondrocytes. The manuscript demonstrated that a TNKS inhibitor significantly suppressed mechanical stress-induced expression of catabolic proteases by inhibition of IL- 1β production through NF-κB and β-catenin-positive interactions, and the expressions of TNKS might be associated with OA progression. The results of the current study are interesting.
Comments on the Quality of English LanguageNoticeable grammatical mistakes were evident in the paper. Please have the paper reviewed for English grammar and style.
Reviewer 2 Report
Comments and Suggestions for Authors
The authors studied the effects of tankyrase inhibitor on human articular chondrocytes after mechanical induced stress.
Comments
1. Abstract: Abbreviation should be disclosed on first use.
2. Lines 31-32: It is not clear why decreased expression of COL2A1, ACAN and SOX9 are considered as anabolic effect? This should be clarified.
3. Abstract: The authors should present a description of specimens and methods used in their study.
4. Lines 65, 300, 301 : The references are required at the end of these sentences.
5. Section 2.1; Lines 328-330: The authors should present evidence that 8% elongation was optimal for the examined cells. They should conduct additional experiment with different percentage of elongation.
6. Fig 1A: The authors should explain why they do not consider the anabolic activity of CTS at 24h. This should be clarified.
7. All the graphs with gene expression assays: At least three Control samples should have been assayed and a standard deviation should be indicated on the graph.
8. In each section and on each Figure caption the authors should indicate which cells they examined.
9. Sections 2.1 and 2.2: It is not clear how the maximum of IL-1b concentration at 24h of incubation was associated with upregulation of anabolic gene expression. Moreover, “3-10% strain led to anabolic responses” (Line 214). This should be clarified.
10. Lines 207-208: This statement was not supported by the experimental data . This should be corrected.
11. Lines 222-224: This explanation is not sufficient/ It should be supported by experimental data.
12. Line 279: Tumor growth factor (TGF) is not presently known. This should be clarified.
13. Line 333: It is not clear why tankyrase inhibitor was used in concentration of 10uM? The authors should try several concentrations of the inhibitor to validate concentration they chose.
14. Line 353: City and Country should be indicated.
15. The Section 4.8: Patients clinical traits should be presented.
16. Conclusions should describe the major authors findings. The first sentence should be moved down.
Reviewer 3 Report
Comments and Suggestions for Authors
The manuscript by Yoshifumi Hotta et al. claims that a TNKS inhibitor significantly suppressed mechanical stress-induced expression of catabolic proteases by inhibition of IL-1β production through NF-κB and β-catenin-positive interactions, and the expressions of TNKS might be associated with OA progression. The results of the current study elucidated the involvement of TNKS in OA pathogenesis and to investigate the effect of TNKS inhibitors on mechanical stress-induced cartilage degeneration, which suggesting a therapeutic role for TNKS inhibitors against OA. However, the manuscript could be improved after revision if the following comments and questions are addressed.
1. In figure1, why were COL2A1, ACAN, and SOX9 levels decreased up to 0-12 h after CTS, whereas, increased in 24h group compared to control group?
2. The important protein expression of TNKS-1/2 level in figure1C and D, and Figure 2C should be added.
3. The WB experiment of nuclear and cytoplasmic extraction of ADAMTS-5 and MMP-13, NF-κB p65 and β-catenin should be added in Figure 4-6.
Comments on the Quality of English Language
The manuscript is understandable, however it is not fluent and the language should be improved.